# Corneal Epithelial Wavefront Error as a Novel Diagnostic Marker for Epithelial Basement Membrane Dystrophy

**DOI:** 10.3390/life14091188

**Published:** 2024-09-20

**Authors:** Vitus Grauvogl, Wolfgang J. Mayer, Jakob Siedlecki, Niklas Mohr, Martin Dirisamer, Siegfried G. Priglinger, Stefan Kassumeh, Nikolaus Luft

**Affiliations:** 1Department of Ophthalmology, UK Augsburg, Stenglinstrasse, 86156 Augsburg, Germany; vitus.grauvogl@uk-augsburg.de; 2Department of Ophthalmology, LMU Klinikum, Mathildenstrasse, 80336 Munich, Germany; wolfgang.j.mayer@med.uni-muenchen.de (W.J.M.); jakob.siedlecki@med.uni-muenchen.de (J.S.); niklas.mohr@med.uni-muenchen.de (N.M.); martin.dirisamer@med.uni-muenchen.de (M.D.); siegfried.priglinger@med.uni-muenchen.de (S.G.P.); stefan.kassumeh@med.uni-muenchen.de (S.K.)

**Keywords:** cornea, dystrophy, treatment lasers, ocular surface

## Abstract

**Synopsis:** Corneal epithelial wavefront error and epithelial thickness variance qualify as highly sensitive and specific biomarkers for epithelial basement membrane dystrophy (EBMD). The biomarkers show a normalization after treatment of EBMD with phototherapeutic keratectomy. **Purpose:** To gauge the diagnostic value of epithelial basement membrane dystrophy (EBMD), a novel spectral-domain optical coherence tomography (SD-OCT)-based imaging modality for simultaneous morphological (thickness profile) and refractive (optical wavefront) assessment of the corneal epithelial layer in one of the most common but often underdiagnosed corneal dystrophies. **Methods:** In this prospective observational study, a total of 32 eyes of 32 patients diagnosed with EBMD and 32 eyes of 32 healthy control subjects were examined with high-resolution anterior segment SD-OCT (MS-39; CSO, Florence, Italy). Various epithelial thickness and epithelial wavefront-derived terms were compared between groups and receiver operating characteristic (ROC) curves were computed to analyze the diagnostic capacity of the respective parameters. A total of 17 of 32 EBMD patients underwent treatment with phototherapeutic keratectomy (PTK) and were followed up for 3 months. **Results:** Epithelial thickness variance (60.4 ± 56.7 µm versus 7.6 ± 6.1 µm) and interquartile range (11.0 ± 6.9 versus 3.3 ± 1.9 µm) were markedly elevated in EBMD patients as compared with healthy controls (both with *p* < 0.001). Epithelial wavefront analysis showed a highly statistically significant excess in all examined aberration terms in EBMD patients (all with *p* < 0.001). Significantly greater areas under the curve (AUCs) were yielded by the epithelial wavefront-derived parameters (e.g., total epithelial wavefront error: AUC = 0.966; 95% confidence interval (CI) 0.932–1) than by the epithelial thickness-derived parameters (e.g., variance: AUC = 0.919; 95% CI 0.848–0.990). **Conclusions**: Corneal epithelial wavefront aberrometry proved valuable as an objective biomarker for EBMD, with high sensitivity and specificity. PTK resulted in a reduction of morphological and refractive epithelial irregularities in EBMD.

## 1. Introduction

The International Classification of Corneal Dystrophies (IC3D), originally published in 2008 and recently revised in 2024 [1,2], defines four broad categories of corneal dystrophies. Classified as category 1, one of the overall most common corneal dystrophies is epithelial basement membrane dystrophy (EBMD). In the pre-IC3D era, it was also commonly referred to as ‘map-dot-fingerprint dystrophy’ due to its characteristic phenotype on slit-lamp examination [3]. The prevalence of EBMD is reported to be as high as 2–7.5% [4,5,6], and an association with the TGFBI gene is assumed [7]. Histologically, redundant sheets of basement membrane material in between distorted epithelium and the intact Bowman’s layer, irregular accumulations of fibrillo-granular material in the subepithelial space, and intraepithelial pseudocysts can be found [2].

Even though highly effective treatment for EBMD is widely available, such as manual superficial keratectomy [8] or Excimer laser-based phototherapeutic keratectomy (PTK) [9,10,11], the condition is frequently underdiagnosed on subjective slit-lamp examination, especially in asymptomatic (“silent”) EBMD. Undiagnosed EBMD can hamper precise intraocular lens power calculation [12] and consequently lead to unsatisfactory visual outcomes after cataract surgery, especially when toric or multifocal intraocular lenses (IOLs) are employed [6,12,13]. Recently, Bellucci et al. reported subpar functional and patient-reported outcomes after regular cataract surgery, as well as the need for unscheduled postoperative visits, in 100% of EBMD patients [14]. Furthermore, missing EBMD on pre-keratorefractive surgery examination can lead to dreadful complications (i.e., epithelial sloughing) during femtosecond-laser-based procedures like femtosecond-laser-assisted laser in situ keratomileusis (fs-LASIK) [15].

Hence, there is an unmet need for objective biomarkers to diagnose EBMD with greater sensitivity and specificity, not only for screening purposes but also for evaluating the efficacy of therapeutic approaches (e.g., PTK). Recently, two studies confirmed that the epithelial thickness irregularities inherent to EBMD can be quantified by anterior segment optical coherence tomography (AS-OCT)-based epithelial thickness mapping with high diagnostic capacity [16,17]. However, Buffault et al. [16] only investigated the standard deviation of epithelial thickness as the only discriminatory marker for EBMD, and their study showed statistical shortcomings regarding the sample size of the control group. Levy et al. [17] confirmed these findings and further indicated an inferior epithelial thickening pattern in most EBMD cases.

Epithelial wavefront aberrometry represents a novel AS-OCT-based analysis of the epithelial layer’s optical wavefront error (WFE). In analogy to corneal wavefront aberrometry, the technique utilizes a ray-tracing approach by tracing a virtual beam of collimated light rays that are deflected by the epithelial layer and both of its interfaces, respectively. An initial study by Canto-Cerdan et al. recently confirmed the reliability of epithelial wavefront aberrometry when analyzing optical power changes due to epithelial thickness remodeling during the early postoperative phase after keratorefractive surgery [18]. The present study is the first to explore the diagnostic value of epithelial wavefront aberrometry in one of the most frequently overlooked corneal diseases—EBMD.

## 2. Materials and Methods

In this prospective observational study performed at the University Eye Hospital of the Ludwig Maximilian University (LMU), consecutive patients presenting with EBMD were included from our institution’s corneal disease clinic. Informed consent was obtained from all participants and all study-related procedures and examinations were conducted in accordance with the ethical standards of the institutional ethics committee of the University Hospital Munich LMU (ethic committee vote number 22-1001) as well as in accordance with the Declaration of Helsinki [19].

The diagnosis of EBMD was in all cases based on thorough slit-lamp examination including fluorescein staining, and was confirmed by one of two consultant-level corneal specialists (N.L. or W.J.M.). Slit-lamp biomicroscopic diagnostic criteria included pathognomonic EBMD phenotypes like maps, dots, fingerprint-like lines, or bleb patterns at the level of Bowman’s layer [1]. Healthy control subjects with normal corneal slit-lamp, fluorescein, and topography exams were recruited from the department’s refractive surgery clinic. Exclusion criteria were any history of preceding corneal surgery as well as contact lens wear during a period of four weeks prior to the corneal measurements, or other ophthalmic disease that might interfere with corneal measurements (e.g., dry eye).

### 2.1. Epithelial Thickness Analysis

All participants underwent high-resolution anterior segment SD-OCT (MS-39; CSO, Florence, Italy). Adequate image acquisition quality was ascertained by the onboard software’s scan quality indicator (“OK”) in all cases. For epithelial thickness analyses, raw data export of epithelial pachymetric data was conducted using the “.csv export” function. The CSV document was then imported into the “RStudio” software for further statistical analysis (RStudio, Version 2023.03.1+446, © 2022 by Posit Software, PBC). The MS-39 enables corneal epithelial thickness mapping over the central 8.00 mm zone. As SD-OCT-based epithelial thickness measurements in the corneal mid-periphery (i.e., 6.0–8.0 mm) are known to show subpar repeatability [20], only the central 6.0 mm zone was analyzed for the purpose of this study. The MS-39 measures epithelial thickness over the central 6.0 mm zone as an array of circular point measurements in 256 circular hemi-meridians with a spacing of 200 µm, resulting in a total of 7680 measurement points per scan. For statistical analysis, the (1) mean epithelial thickness, (2) minimum, (3) maximum, (4) interquartile range (IQR), and (5) variance for each scan were calculated and compared between groups.

### 2.2. Epithelial Wavefront Analysis

The MS-39 analyzes the epithelium’s optical wavefront as determined by the simulation of light interaction with the epithelial layer and its interfaces, the air–tear film interface and the epithelium–stroma interface. Utilizing a ray-tracing approach, a bundle of collimated rays, possessing a predetermined diameter, is virtually projected onto the epithelium. These rays are then deflected based on the corneal slope and subsequently adjusted according to the slope of Bowman’s layer. Based on the virtually refracted rays, different components of the epithelial wavefront errors (WFEs) are computed as root mean square (RMS) values, which are commonly used in ocular wavefront aberrometry [18]. For the purpose of this study, the refractive index of the epithelium was consistently set to 1.401 and the epithelial wavefront analysis was centered on the corneal vertex with a virtual pupil diameter of 6.00 mm [18]. Using the onboard software, the following epithelial wavefront aberration terms were extracted for statistical analysis: (1) spherical WFE, (2) cylindrical WFE, (3) spherical aberration RMS, (4) coma RMS, (5) trefoil RMS, (6) total higher order aberrations (HOAs) RMS, (7) total WFE RMS and (8) residual WFE RMS (pre-defined by the onboard software as the total WFE RMS excluding the terms astigmatism, coma, and spherical aberration).

### 2.3. Phototherapeutic Keratectomy

For a total of 17 of 32 EBMD patients, additional data were collected as part of a follow-up visit three months after phototherapeutic keratectomy (PTK). Standard epithelium-off PTK was performed under topical anesthesia (tetracain hydrochloride 1% eye drops, Pharmacy of LMU Klinikum, Munich, Germany) with a standardized treatment zone of 8.00 mm and a programmed ablation depth of 12 µm, using the MEL90 Excimer platform (Carl Zeiss Meditec AG, Jena, Germany).

### 2.4. Clinical Examination

In addition to standard slit-lamp microscopy with fluorescein staining, all subjects underwent subjective manifest refraction testing, as determined by the Jackson cross-cylinder, using standard ETDRS test charts at a 4 m distance. Best-corrected visual acuity (BCVA) was analyzed in logMAR (logarithm of the minimum angle of resolution). In order not to interfere with SD-OCT imaging, standard Goldmann applanation tonometry was performed as the last examination during each visit.

### 2.5. Statistical Analysis

Statistical analysis was performed using “RStudio”. The Kolmogorov–Smirnov test was used to assess the normality of the data. The differences between the EBMD group before and after PTK and the healthy control group were analyzed using Fisher’s F-test, the Welch test, and Student’s *t*-test. The Mann–Whitney U-test was employed to compare parameters between the EBMD group and the healthy control group. Receiver operating characteristic (ROC) curves and corresponding areas under the curve (AUCs) with 95% confidence intervals (95% CIs) were computed to determine the diagnostic capacity of the various epithelial parameters between EBMD patients and controls. ROC curves were tested for statistical significance following the Mason and Graham method [21]. Optimal cutoff values were calculated using the Youden Index [22]. A significance level of *p* < 0.05 was defined as an indicator of statistical significance.

## 3. Results

The mean age was comparable between the EBMD group (51 ± 17 years; range 27 to 87) and the healthy control group (46 ± 14 years; range 31 to 79; *p* = 0.30), with an identical female to male ratio of 17:15 in both groups. Best corrected visual acuity (BCVA) was better in the healthy control group (0.01 ± 0.56 logMAR; range −0.15 to 0.49) compared to the EBMD group (0.15 ± 0.50 logMAR; range −0.10 to 0.60; *p* < 0.01). Manifest refraction was comparable between groups regarding the spherical equivalent (EBMD: −2.54 ± 2.61 diopters; D; range −7.50 to +3.13; healthy controls: −2.93 ± 3.32 D; range −9.50 to +3.88 D; *p* = 0.31) and the cylindrical component of refraction (EBMD −1.60 ± 1.71 D; range −8.00 to −0.25; healthy controls: −1.20 ± 1.21 D; range −5.75 to −0.25; *p* = 0.33). Intraocular pressures in the EBMD group (17.3 ± 2.4 mmHg; range 13 to 22) and the healthy group (15.8 ± 3.4 mmHg; range 11 to 26; *p* = 0.09) were comparable.

Table 1 summarizes the epithelial thickness-derived parameters for the EBMD group and healthy controls. The epithelial layer was generally thicker (*p* < 0.001) (see Figure 1A) and more inhomogeneously distributed in EBMD patients. As an indicator of epithelial thickness irregularity, epithelial thickness variance in the EBMD group (60 ± 57 µm^2^) was statistically significantly elevated when compared with that in healthy controls (8 ± 6 µm^2^; *p* < 0.001) (Figure 1E). Accordingly, the interquartile range (IQR) was also statistically significantly higher in EBMD patients (11 ± 7 μm versus 3 ± 2 μm; *p* < 0.001; Figure 1D). As a further indicator of greater inhomogeneity of the epithelial thickness profile in EBMD, the minimum epithelial thickness was lower (*p* = 0.03) (Figure 1B), while the maximum epithelial thickness was higher in EBMD eyes (*p* < 0.001) as compared with healthy controls (Figure 1C).

In the ROC analysis for epithelial thickness-derived parameters, epithelial thickness variance showed the best discriminatory power with an AUC of 0.919 (95% CI 0.848–0.990), which was statistically significant (*p* = 0.04; Figure 2A). The second-best discriminatory parameter was the IQR with an AUC of 0.895 (95% CI 0.183–0.976; *p* < 0.01; Figure 2B), followed by the maximum epithelial thickness (AUC 0.878; 95% CI 0.786–0.968, *p* < 0.001) and minimal epithelial thickness (AUC 0.678; 95% CI 0.544–0.81; *p* = 0.01). The discriminatory capacity of the mean epithelial thickness (AUC 0.767; 95% CI: 0.650–0.884; *p* = 0.07) failed the predefined level of statistical significance according to the Mason and Graham method [21]. The optimal cutoff values for discriminating between healthy and EBMD eyes were an epithelial thickness variance of 16 µm (sensitivity 0.84 and specificity 0.88), an epithelial thickness IQR of 7 µm (sensitivity 0.75 and specificity 0.94), a maximum epithelial thickness of 62 µm (sensitivity 0.81 and specificity 0.88), and a minimal epithelial thickness of 39 µm (sensitivity 0.88 and specificity 0.47).

Table 2 summarizes the epithelial wavefront-derived parameters for the EBMD group and healthy controls. All analyzed higher-order aberration (HOA) terms were statistically significantly higher in the EBMD group than in the healthy control group (all with *p* < 0.001). Both sphere and cylinder showed statistically significantly more negative values in the EBMD group (<0.001).

Pertaining to the ROC analyses for epithelial wavefront-derived parameters, the total WFE RMS yielded the highest AUC of 0.975 (95% CI 0.946–1 µm; Figure 3A), followed by the residual WFE RMS (AUC 0.954; 95%-CI 0.910–0.999 µm; Figure 3B) and total HOAs RMS (AUC 0.950; 95%-CI 0.897–1; Figure 3C), all with *p* < 0.001. All three parameters surpassed the diagnostic capacity of the most discriminative epithelial thickness-derived parameter, epithelial thickness variance (AUC 0.919). The remainder of the analyzed epithelial wavefront parameters showed lower AUC values than epithelial thickness variance. Coma RMS yielded an AUC of 0.872 (95%CI: 0.784–0.963 µm; *p* < 0.001), spherical WFE had an AUC of 0.858 (95% CI: 0.731–0.985 µm; *p* < 0.001), cylindrical WFE showed an AUC of 0.780 with a 95% CI of 0.668–0.892 µm (*p* < 0.001), and spherical aberration had an AUC of 0.759 with a 95% CI of 0.632–0.885 µm (*p* = 0.001). The diagnostic capacity of trefoil did not meet the predefined level of statistical significance (AUC 0.595; 95% CI 0.388–0.803 µm; *p* = 0.36). The optimal wavefront analysis-derived cutoff values for discriminating between healthy and EBMD eyes were a total WFE RMS of 1.04 µm (sensitivity = 0.83 and specificity = 1), a residual WFE RMS of 0.36 µm (sensitivity = 0.90 and specificity = 0.87), a total HOAs RMS of 0.54 µm (sensitivity = 0.87 and specificity = 1), a coma RMS of 0.34 µm (sensitivity = 0.67 and specificity = 0.97), a spherical WFE RMS of −0.49 D (sensitivity = 0.97 and specificity = 0.60), a cylindrical WFE RMS of −0.56 D (sensitivity = 0.92 and specificity = 0.67), and a spherical aberration RMS of 0.23 µm (sensitivity = 0.53 and specificity = 1).

A total of 17 (53%) of 32 patients were available for a follow-up examination three months after undergoing PTK. Epithelial thickness analysis showed statistically significant thinning and regularization of the epithelial thickness profile due to PTK (Table 3). Mean epithelial thickness decreased statistically significantly, as did variance and IQR (Figure 1A,D,E), the latter two indicating a more homogeneous thickness distribution of the epithelial layer. The minimum epithelial thickness increased (*p* = 0.02) while the maximum epithelial thickness decreased, though not to a statistically significant extent (Figure 1C). As illustrated in Figure 1D, the IQR (*p* < 0.01) decreased to a level that was statistically equivocal to healthy control subjects (*p* = 0.05). PTK also induced a statistically significant decrease of epithelial thickness variance (*p* < 0.01), however, not to the level of healthy controls (Figure 1E). In concordance with this morphological regularization, the optical properties of the epithelial layer also improved after PTK (Table 4). Remarkably, as visualized in Figure 4 and Figure 5, PTK led to an improvement in the majority of the higher-order aberration terms. Nevertheless, the levels of the respective HOA terms did not approximate those of healthy controls as closely as did the thickness-derived parameters after PTK. Only the lower-order aberration (LOA) term spherical WFE RMS (Figure 4A) showed a normalization after PTK (with *p* < 0.001) to levels that were statistically comparable with healthy controls.

## 4. Discussion

The present study introduces epithelial wavefront aberrometry as a novel biomarker for EBMD with higher diagnostic accuracy than epithelial thickness mapping. This AS-OCT-based imaging tool utilizes a ray-tracing approach by tracing a virtual beam of collimated light rays that are deflected by the epithelial layer and both of its interfaces, respectively. A recent study by Canto-Cerdan confirmed the reliability of epithelial wavefront aberrometry for analyzing epithelial power changes after keratorefractive surgery [18]. The present study is the first to propose its diagnostic utility as a biomarker in corneal disease.

It is well known from previous AS-OCT-based studies that the epithelial thickness profile is significantly altered in EBMD [16]. For instance, Buffault et al. reported a highly irregular corneal epithelium in EBMD patients. Specifically, the EBMD-affected epithelium showed a propensity towards thickening at the inferior hemisphere of the cornea [16]. These findings were also confirmed by Levy et al. in their 2022 study [17]. Gravity, regular blinking, and shearing of the eyelids as well as a weaker connection between the epithelium and its basement membrane are suspected to cause the epithelium to be pushed downwards and, consequently, agglomerate at the inferior aspect of the cornea. In an effort to utilize these irregularities diagnostically, previous groups proposed the standard deviation (SD) of the corneal epithelial thickness as an objective measure for epithelial thickness irregularity. Using the SD of epithelial thickness in ROC analyses yielded areas under the curve of 0.90 [16] and 0.97 [17] for the discrimination between EBMD and non-EBMD eyes, respectively. These results are consistent with our findings regarding the variance and interquartile range of epithelial thickness (AUCs of 0.92 and 0.89, respectively). Also, previous findings of a lowered minimum epithelial thickness and an elevated maximum epithelial thickness in eyes affected by EBMD were confirmed by the present study [17].

Due to the irregularity of the epithelium’s thickness profile, its optical quality deteriorates, as does patients’ subjective quality of vision [14]. A recent study by Bellucci et al. reported the subpar optical properties of EBMD corneas, analyzing the objective scattering index (OSI), point spread function (PSF), and the modulation transfer function (MTF) [14]. It is important to note that the authors analyzed ocular aberrations, unlike our research. In the present study, we specifically analyzed the optical properties of the epithelial layer in EBMD in an effort to screen for potential diagnostic disease markers. Of all investigated epithelial wavefront-derived parameters, the ‘Total WFE RMS’ showed the best discriminatory power between healthy and EBMD patients (AUC = 0.975), followed by the ‘Residual WFE RMS’ (AUC = 0.954) and ‘Total HOAs RMS’ (AUC = 0.950). All of these parameters outperformed the best epithelial thickness profile-derived “irregularity indices” variance and IQR. In a clinical setting, the authors suggest the use of the novel parameter ‘Total WFE RMS’ with the cutoff set to 1.04 µm, which offered high diagnostic capability in this study (sensitivity = 0.83 and specificity = 1). Clinical applications of this easy-to-use parameter could range from screening purposes prior to cataract or corneal refractive surgery, disease grading and monitoring, or evaluating the efficacy of therapeutic interventions [23].

A supplementary finding of this study was that the irregularity of the EBMD-affected corneal epithelium decreased to almost healthy levels after treatment with PTK. As shown in Figure 3, the collective suffering from EBMD showed a remarkable reduction in epithelial thickness variance, from 60 µm to 18 µm, whereas healthy eyes ranged at 8 µm. Analysis of the epithelial thickness IQR yielded similar results. The regularization of the epithelium was accompanied by a reduction in epithelial LOAs (i.e., spherical and cylindrical WFE, respectively). Previous studies have delivered inconclusive refractive outcomes of PTK in EBMD, which can result in myopization, hyperopization, or no change in refraction at all [9,24,25]. In our study, we observed an emmetropization of the myopic epithelial WFE after PTK, resulting in a hyperopic “shift” of 0.78 D at the corneal plane. Hence, we hypothesized that this refractive shift does not arise from the 12 µm ablation of Bowman’s layer and stroma, but from the normalization of the epithelium’s spherical wavefront error to normal levels due to PTK. Accordingly, a potential explanation for the conflicting previous findings concerning the refractive effect of PTK [9,24,25] may lie in the differences of the EBMD-induced epithelial wavefront error between the study cohorts. Nevertheless, further research is required to further substantiate this hypothesis.

As opposed to this regularization of the epithelial LOA profile (spherical and cylindrical WFE), epithelial HOAs experienced a statistically significant decrease, albeit not to levels that were statistically comparable with the healthy cohort. For instance, ‘Coma RMS’ decreased from 0.53 µm to 0.32 µm (*p* < 0.03) but was still elevated when compared with the healthy cohort (0.27 µm; *p* < 0.001). These results are comparable with findings by Yildiz et al., who investigated corneal HOAs of patients undergoing PTK-treatment for corneal subepithelial infiltrates after epidemic keratoconjunctivitis [26]. One possible reason for the improved but not normalized HOA levels might be the incomplete or late regeneration of the ablated Bowman’s layer, which lies directly posterior to the epithelial basement membrane (EBM) and anterior to the corneal stroma. The acellular Bowman’s layer is composed of randomly-oriented collagen fibrils and has a thickness of 8–12 µm in humans [27]. Currently, Bowman’s layer is considered non-regenerating after PTK by some entities [28]. Others have found that a Bowman’s-like layer can regenerate beneath the epithelial basement membrane ten years after Excimer-laser ablation [27].

This research was conducted exclusively using the MS-39 device; however, the parameters we utilized, such as those central to our methodology, are in principle also measurable with other devices, such as the ANTERION (©Heidelberg Engineering, Heidelberg, Germany). Previous studies have shown that morphological epithelial measurements (i.e., epithelial thickness mapping) from these two devices are generally comparable, though not interchangeable [29]. Nonetheless, it is important to acknowledge that the optical analysis of the epithelial wavefront (i.e., epithelial wavefront aberrometry) is, to the best of our knowledge, as of today exclusively available with the MS-39 platform.

An interesting avenue for future research could be the deeper analysis of diseased corneas, particularly with respect to the severity of the disease (i.e., disease grading). The severity of EBMD may be significant for both diagnostics and treatment. We hypothesize that a subclinical stage of EBMD may already exhibit elevated aberrometry levels, although this has yet to be proven. Further research is required to assess the role of these parameters in disease progression over time, especially as EBMD can also disappear at intervals and after treatment.

This pilot study is primarily limited by its small sample size. Moreover, not all the EBMD patients were available for a post-PTK examination visit, which might be interpreted as a selection bias. In addition, even though the follow-up time after PTK of three months may be considered sufficient for epithelial basement membrane regeneration and epithelial remodeling, as indicated above, longer regeneration times may be expected for a Bowman’s-like layer [27].

In conclusion, we propose corneal epithelial wavefront aberrometry as a novel, objective, non-invasive diagnostic test for EBMD. The ‘Total Epithelial WFE’ showed the highest diagnostic capability of all examined parameters and may serve clinicians as a valuable adjunct in diagnosing EBMD.

## 5. Conclusions


**What is already known on this topic**



*EBMD is diagnosed clinically by slit-lamp examination, but the search for a suitable diagnostic test is still ongoing. PTK treatment offers a reliable and safe treatment option for patients suffering from EBMD, though there is little evidence for an actual smoothing effect on the epithelial profile.*



**What this study adds:**

*This study introduces a novel and easy-to-use biomarker suitable for discrimination between healthy and EBMD eyes: the epithelial wavefront error.*

*This study provides evidence of smoothing of the corneal epithelium and reduction of the epithelial wavefront error after PTK.*




**How this study might affect research, practice or policy**



*Analysis of the corneal epithelium using novel OCT-based methods such as epithelial mapping and epithelial aberrometry can assist clinicians in diagnosing EBMD and may be suitable for screening examinations, e.g., before cataract surgery.*


## Figures and Tables

**Figure 1 life-14-01188-f001:**
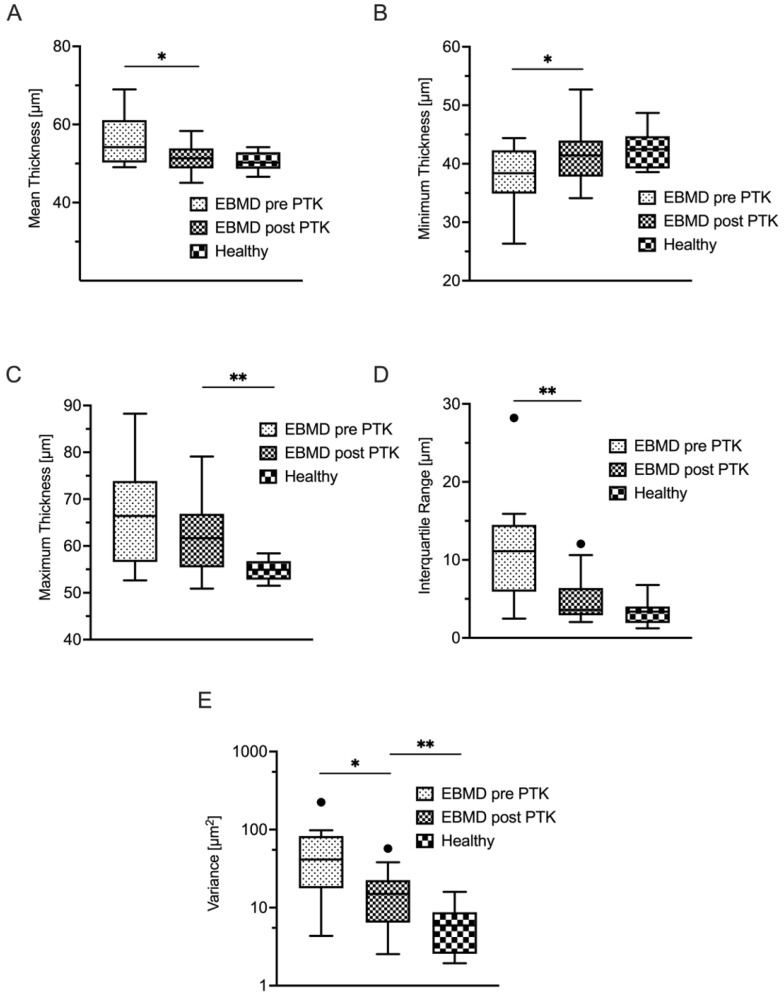
Terms of mean (**A**), minimum (**B**) and maximum (**C**) epithelial thickness as well as interquartile range (**D**) and variance (**E**) of epithelial thickness. The reduction of IQR after PTK showed the highest significance, followed by variance, mean thickness, and minimum thickness. The maximum epithelial thickness did not show a statistically significant decrease after PTK. * *p* < 0.05; ** *p* < 0.01; EBMD, epithelial basement membrane dystrophy; PTK, phototherapeutic keratectomy.

**Figure 2 life-14-01188-f002:**
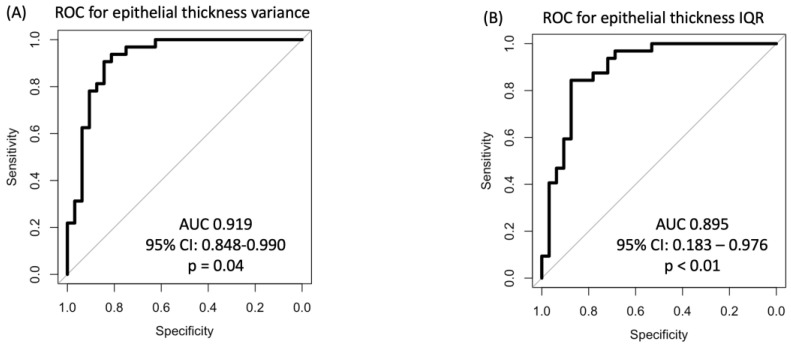
ROC analyses of (**A**) the variance of epithelial thickness and (**B**) the IQR show a high capacity for discrimination between healthy and EBMD corneas. AUC, area under the curve; EBMD, epithelial basement membrane dystrophy; IQR, interquartile range; ROC, receiver operating characteristic.

**Figure 3 life-14-01188-f003:**
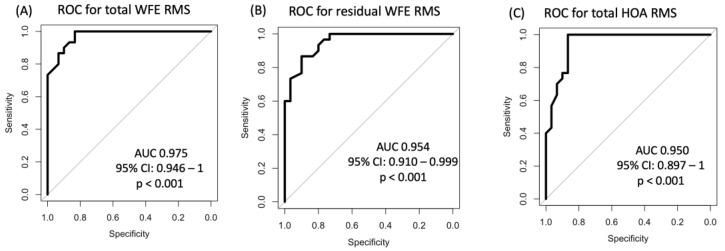
ROC analyses of the wavefront errors between healthy individuals and EBMD patients prove an even higher discriminatory power compared to Figure 2. The highest AUC was achieved for analysis of the total WFE RMS (**A**), followed by the residual WFE RMS (**B**) and total HOA RMS (**C**). ROC, receiver operating characteristic; WFE, wave front error; EBMD, epithelial basement membrane dystrophy; HOA, higher order aberrations.

**Figure 4 life-14-01188-f004:**
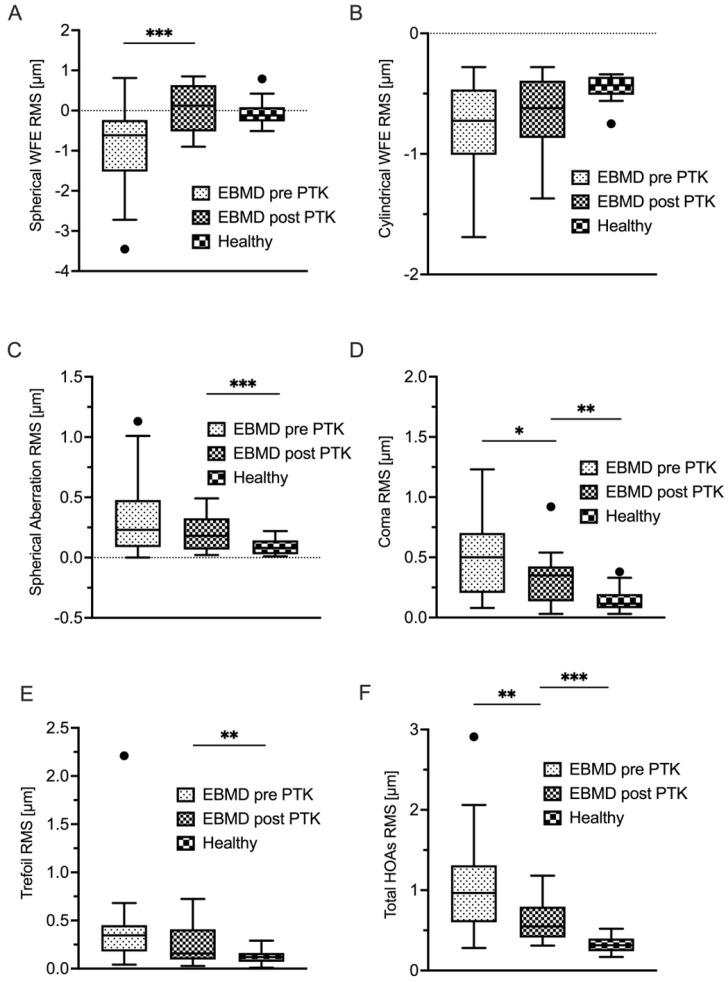
Corneal epithelial aberrometry. Displayed are lower-order aberrations spherical WFE (**A**) und cylindrical WFE (**B**) and higher-order aberrations spherical aberration (**C**), coma (**D**), trefoil (**E**) and total HOA (**F**) in EBMD patients before and after PTK, as well as in a healthy control group. * *p* < 0.05; ** *p* < 0.01; *** *p* < 0.001; WFE, wavefront error; EBMD, epithelial basement membrane dystrophy; PTK, phototherapeutic keratectomy; RMS, root mean square.

**Figure 5 life-14-01188-f005:**
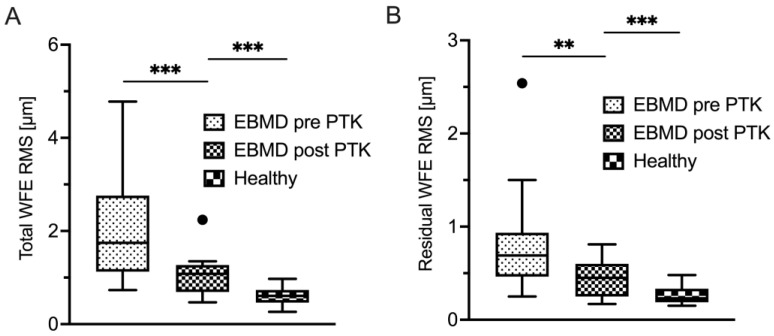
Corneal epithelial aberrometry—summative terms. Displayed are the total WFE (**A**), as well as residual WFE (**B**), in EBMD patients before and after PTK as well as in a healthy control group. * *p* < 0.05; ** *p* < 0.01; *** *p* < 0.001; WFE, wavefront error; EBMD, epithelial basement membrane dystrophy; PTK, phototherapeutic keratectomy; RMS, root mean square.

**Table 1 life-14-01188-t001:** Epithelial thickness-derived parameters in the EBMD versus control groups.

Epithelial Thickness	EBMD (n = 32)	Control Group (n = 32)	*p*-Value *
Mean (µm)	58 ± 9 Range 46 to 87	51 ± 3 Range 45 to 60	<0.001
Min (µm)	40 ± 8 Range 26 to 64	43 ± 5 Range 35 to 56	0.03
Max (µm)	70 ± 12 Range 53 to 101	57 ± 5 Range 52 to 74	<0.001
IQR (µm)	11 ± 7 Range 1 to 28	3 ± 2 Range 1 to 10	<0.001
Variance (µm^2^)	60 ± 57 Range 3 to 224	8 ± 6 Range 1 to 29	<0.001

EBMD, epithelial basement membrane dystrophy; IQR, interquartile range; * Welch test.

**Table 2 life-14-01188-t002:** Epithelial wavefront-derived parameters in the EBMD vs. control groups.

WFE Components	EBMD (n = 32)	Control Group (n = 32)	*p*-Value *
Spherical WFE (D)	−0.85 ± 1.00 Range −3.45 to 0.81	−0.07 ± 0.31 Range −0.51 to 0.79	<0.001
Cylindrical WFE (D)	−0.75 ± 0.36 Range −1.69 to −0.28	−0.45 ± 0.11 Range −0.75 to −0.34	<0.001
Spherical aberration RMS (µm)	0.32 ± 0.30 Range 0.00 to 1.13	0.09 ± 0.06 Range 0.01 to 0.22	<0.001
Coma RMS (µm)	0.55 ± 0.36 Range 0.08 to 1.23	0.14 ± 0.09 Range 0.02 to 0.38	<0.001
Trefoil RMS (µm)	0.38 ± 0.38 Range 0.04 to 2.21	0.12 ± 0.06 Range 0.01 to 0.29	<0.001
Total HOAs RMS (µm)	1.12 ± 0.63 Range 0.28 to 2.91	0.33 ± 0.10 Range 0.17 to 0.52	<0.001
Total WFE RMS (µm)	2.14 ± 1.20 Range 0.73 to 4.78	0.62 ± 0.20 Range 0.26 to 1.07	<0.001
Residual WFE RMS (µm)	0.88 ± 0.66 Range 0.25 to 3.49	0.27 ± 0.09 Range 0.15 to 0.48	<0.001

WFE, wavefront error; EBMD, epithelial basement membrane dystrophy; D, diopters; HOAs, higher order aberrations; RMS, root mean square; * Welch test.

**Table 3 life-14-01188-t003:** Epithelial thickness parameters in the EBMD group pre-PTK vs. post-PTK.

Epithelial Thickness	Pre-PTK (n = 17)	Post-PTK (n = 17)	*p*-Value *
Mean (µm)	56 ± 7 Range 49 to 69	52 ± 4 Range 45 to 58	0.03
Min (µm)	38 ± 5 Range 26 to 44	42 ± 5 Range 35–53	0.02
Max (µm)	67 ± 11 Range 53 to 88	62 ± 8 Range 51 to 79	0.10
IQR (µm)	11 ± 7 Range 3 to 28	5 ± 3 Range 2 to 12	<0.01
Variance (µm^2^)	56 ± 54 Range 4 to 224	18 ± 15 Range 3 to 57	0.01

PTK, phototherapeutic keratectomy; IQR, interquartile range; * paired *t*-test.

**Table 4 life-14-01188-t004:** Epithelial WFE parameters in the EBMD group pre-PTK vs. post-PTK.

WFE Components	Pre-PTK (n = 17)	Post-PTK (n = 17)	*p*-Value *
Spherical WFE (D)	−1.00 ± 0.96 Range −3.61 to −0.10	0.07 ± 0.57 Range −0.90 to 0.85	<0.001
Cylindrical WFE (D)	0.69 ± 0.38 Range −1.31 to −0.28	−0.68 ± 0.34 Range −1.37 to −0.28	0.36
Spherical aberration RMS (µm)	0.29 ± 0.34 Range 0.00 to 1.13	0.20 ± 0.16 Range 0.02 to 0.49	0.14
Coma RMS (µm)	0.53 ± 0.39 Range 0.08 to 1.23	0.32 ± 0.22 Range 0.03 to 0.92	0.03
Trefoil RMS (µm)	0.30 ± 0.15 Range 0.04 to 0.55	0.25 ± 0.20 Range 0.03 to 0.72	0.11
Total HOAs RMS (µm)	0.62 ± 0.51 Range 0.28 to 2.06	0.63 ± 0.26 Range 0.31 to 1.18	<0.01
Total WFE RMS (µm)	2.04 ± 1.17 Range 0.73 to 4.70	1.07 ± 0.41 Range 0.47 to 2.24	<0.001
Residual WFE RMS (µm)	0.72 ± 0.35 Range 0.25 to 1.50	0.46 ± 0.20 Range 0.17 to 0.81	<0.01

WFE, wavefront error; EBMD, epithelial basement membrane dystrophy; D, diopters; HOAs, higher order aberrations; RMS, root mean square; * Welch test.

## Data Availability

Due to ongoing research, the data is available upon request.

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
