# Peer review of "Corneal Epithelial Wavefront Error as a Novel Diagnostic Marker for Epithelial Basement Membrane Dystrophy"

_life, 2024, doi:10.3390/life14091188_

Round 1
Reviewer 1 Report
Comments and Suggestions for Authors
I would like to begin by congratulating the authors for the originality of the work and also because they are doing research in an area where, as they state in the introduction, there is an unmet need to assess EBMD objectively.
I would like to suggest some minor changes that are intended to increase the academic value of your study.
1. In methods, when you explain that two corneal experts have made the diagnoses, based on the biomicroscopic characteristics of the disease, add a corresponding citation, which supports this sentence:
"Slit-lamp biomicroscopic diagnostic criteria included pathognomonic EBMD phenotypes like maps, dots, fingerprint-like lines or bleb patterns at the level of Bowman’s layer."
2. In methods, concerning your population, just as an observation: it could be interesting to know whether these are cases whose diagnosis is new, or whether they are patients with previously known pathology and are cases of recurrent alterations. This is related to the age of the pathology, which may determine another type of pattern. This, together with the diagnostic tool that you are developing, could be very interesting for follow-up and to know if there are differences in the evolution of the disease. Specifically, my comment/question: can you add the data of the age of the pathology of the cases you have included? If not, nothing happens. But if you can do it, you will be providing other interesting clinical data, for you and future studies that can compare them.
3. Describing your case population a little more, not only with sex and age, but also with the age of the pathology, is one more piece of information to provide reproducibility and make your study more comparable in the future. If you cannot add it in methods, perhaps I suggest you make a brief comment in the discussion. Just a comment, it is not a limitation, but it is interesting that you, as authors, point out that there may be differences and that precisely what you have developed could also be very useful to discern the age and severity of the pathology, perhaps even for objective staging of the disease.
4. Finally, another question/comment, which perhaps you could bring up in the discussion: how dependent is this technique on the OCT model you have used? I understand a priori, that your method is highly dependent on the device used. Is it so? Do you think this technique could be reproducible with another similar SD-OCT? Could you comment on something in the discussion?
Thank you very much
Author Response
We would like to thank the reviewer for his/her/their thoughtful comments and suggestions, which have greatly contributed to improving our manuscript. We have carefully considered each point raised and have addressed them in the revised version of the paper. Below, we provide detailed responses to the reviewers' comments and explain the changes made to the manuscript accordingly.
Comment 1: In methods, when you explain that two corneal experts have made the diagnoses, based on the biomicroscopic characteristics of the disease, add a corresponding citation, which supports this sentence: "Slit-lamp biomicroscopic diagnostic criteria included pathognomonic EBMD phenotypes like maps, dots, fingerprint-like lines or bleb patterns at the level of Bowman’s layer."
Response 1: The corresponding citation linking to the latest IC3D-classification, which includes these diagnostic criteria, was added.
Comment 2: In methods, concerning your population, just as an observation: it could be interesting to know whether these are cases whose diagnosis is new, or whether they are patients with previously known pathology and are cases of recurrent alterations. This is related to the age of the pathology, which may determine another type of pattern. This, together with the diagnostic tool that you are developing, could be very interesting for follow-up and to know if there are differences in the evolution of the disease. Specifically, my comment/question: can you add the data of the age of the pathology of the cases you have included? If not, nothing happens. But if you can do it, you will be providing other interesting clinical data, for you and future studies that can compare them.
Response 2: We would like to thank the reviewer for this very inspirational point he is making. Unfortunately, these data werde not acquired. Since the patients undergoing surgery in our cornea centre are usually referred by an ophalmologist in practice, a clear starting point of the disease is hard to define if not specifically adressed in the anamnesis or in a exchange with the referring colleague. Nevertheless, we will build on the reviewer’s idea by collecting prospective long-term follow-up data on EBMD patients to assess the value of epithelial wavefront aberrometry in longitudinal monitoring of disease progression.
Comment 3: Describing your case population a little more, not only with sex and age, but also with the age of the pathology, is one more piece of information to provide reproducibility and make your study more comparable in the future. If you cannot add it in methods, perhaps I suggest you make a brief comment in the discussion. Just a comment, it is not a limitation, but it is interesting that you, as authors, point out that there may be differences and that precisely what you have developed could also be very useful to discern the age and severity of the pathology, perhaps even for objective staging of the disease.
Response 3: As we think this might be an especially interesting field of research, we added the following paragraph in the manuscript:
An interesting avenue for future research could be the deeper analysis of diseased corneas, particularly with respect to the severity of the disease (i.e. disease grading). The severity of EBMD may be significant for both diagnostics and treatment. We hypothesize that a subclinical stage of EBMD may already exhibit elevated aberrometry levels, although this has yet to be proven. Further research is required to assess the role of these parameters in disease progression over time.
Comment 4: Finally, another question/comment, which perhaps you could bring up in the discussion: how dependent is this technique on the OCT model you have used? I understand a priori, that your method is highly dependent on the device used. Is it so? Do you think this technique could be reproducible with another similar SD-OCT? Could you comment on something in the discussion?
Response 4: Thank you for your insightful question regarding the dependency of our technique on the specific OCT model used. We included the following paragraph in our manuscript to adress the different SD-OCT devices:
This research was conducted exclusively using the MS-39 device; however, the parameters we utilized, such as those central to our methodology, are in principle also measurable with other devices, such as the ANTERION (©Heidelberg Engineering, Heidelberg, Germany). Previous studies have shown that morphological epithelial measurements (i.e. epithelial thickness mapping) from these two devices are generally comparable, though not interchangeable [29]. Nonetheless, it is important to acknowledge that the optical analysis of the epithelial wavefront (i.e. epithelial wavefront aberrometry) is - to the best of our knowledge - as of today exclusively available with the MS-39 platform.

Reviewer 2 Report
Comments and Suggestions for Authors
This paper tries to compute corneal epithelium parameters in EBMD eyes. The authors found irregularity of the epithelium thickness, and high aberration of the corneal epithelium. They conclude that a specific level of aberration can be used to characterize EBMD.
The paper is sound and the data well presented, however the discussion can be improved. While the sensitivity of their approach cannot be questioned, specificity cannot be assumed to be as good as the authors conclude. Therefore in my opinion the new indicator is more valid to set the current level of EBMD in a given eye than to indicate the presence or not of EBMD. In addition EBMD can almost disappear at intervals and after treatment, and EBMD eyes measured duting remission might fail to present the level of aberration required for the diagnosis.
A minor observation: The study by Bellucci et al. (ref 14) employed a double-pass aberrometer and therefore the ocular (and not corneal) aberration was considered. This should be clear in the discussion.
Comments on the Quality of English LanguageQuality of English acceptable. Needs minor changes
Author Response
Dear Reviewer,
Thank you for your valuable feedback and suggestions. In response to your comments, we have made the following additions to the manuscript:
-
We included the following paragraph in the discussion:
"An interesting avenue for future research could be the utility of epithelial wavefront aberrometry for EBMD grading. The severity of EBMD may be significant for both diagnostics and treatment. We hypothesize that mild or clinically subclinical stages of EBMD may already exhibit elevated aberrometry levels, although this has yet to be proven. Further research is also required to assess the role of these novel biomarkers in monitoring of disease progression over time, especially as EBMD can also disappear at intervals and after treatment."
-
To address your point on aberration analysis, we also added the following sentence for clarification:
"It is important to note that the authors analyzed ocular aberrations, unlike our research. In the present study, we specifically analyzed the optical properties of the epithelial layer in EBMD (...)"
We appreciate your insightful comments, which have helped us improve the clarity and depth of our manuscript.
